# Response surface optimization for cadmium biosorption onto the pre-treated biomass of red algae *Digenia simplex* as a sustainable indigenous biosorbent

Sedky H.A. Hassan[1], Maryam M. Alomran[2], Nada I.A. Alsugiran[2], Mostafa Koutb[3], Hassan Ahmed[4] and Mustafa A. Fawzy[4,5]

[1] Department of Biology, College of Science, Sultan Qaboos University, Muscat, Oman
[2] Department of Biology, College of Science, Princess Nourah bint Abdulrahman University, Riyadh, Saudi Arabia
[3] Department of Biology, Faculty of Science, Umm Al-Qura University, Makkah, Saudi Arabia
[4] Faculty of Science, Biology Department, Taif University, Taif, Saudi Arabia
[5] Botany and Microbiology Department, Faculty of Science, Assiut University, Assiut, Egypt

Corresponding author
Maryam M. Alomran, mmalomran@pnu.edu.sa

## ABSTRACT

**Background.** Cadmium pollution from industrial effluent can cause major health concerns, so it must be removed from wastewater prior to disposal. The objective of this study was to remove cadmium ($Cd^{2+}$) from aquatic environments using red macroalgae *Digenia simplex* pretreated with calcium chloride ($CaCl_2$) (DSC).

**Methods.** Batch adsorption studies were carried out to evaluate the individual impacts of adsorbent-metal contact time, cadmium concentration, and temperature on the cadmium removal efficiency and biosorption capacity. The Box-Benhken experimental design of response surface methodology was also used to investigate the relationship between different factors (pH, $Cd^{2+}$ concentration and algal dose) and the cadmium removal efficiency of pretreated *D. simplex*.

**Results.** The highest removal efficiency of 97.27% was achieved by combining different optimal parameters, including pH 5.78, initial $Cd^{2+}$ concentration of 24.79 mg/L, and adsorbent dosage of 6.13 g/L. Moreover, cadmium removal from agricultural wastewater samples by pretreated *D. simplex* was evaluated under the optimal conditions, and the removal rate excessed 97%. Kinetic and isotherm investigations showed that the pseudo-second-order, Freundlich, Langmuir, and Dubinin–Radushkevich models of cadmium biosorption on pretreated algal biomass correlated well with the experimental biosorption data, implying that the biosorption of $Cd^{2+}$ is a homogeneous monolayer and multilayer chemisorption process. The equilibrium isotherm data indicated that the biosorption capacity of the biosorbent was 11.16 mg/g as determined by the Langmuir model. Furthermore, the biosorption process was evaluated as an endothermic process with entropy and enthalpy values of 0.134 kJ/mol K and 38.01 kJ/mol, respectively. The functional groups, surface morphology, and elemental composition of the algal biomass were investigated, revealing the porous nature of the cell surface and the abundance of functional groups responsible for the $Cd^{2+}$ biosorption process. These results suggest that DSC biomass can be used as a biosorbent for the effective removal of $Cd^{2+}$ ions from effluent due to its availability and strong biosorption capability.

## INTRODUCTION

With the acceleration of urbanization and the widespread development of industrial activities, the pollution of aquatic environments by heavy metals has become the main cause of environmental contamination in surface water and groundwater (*Singh et al., 2024*). Heavy metals are persistent, non-biodegradable pollutants that build up in the environment and contaminate the food chain. Heavy metal accumulation in living organisms poses a risk to human and animal health (*Nowicka, 2022*).

Cadmium ($Cd^{2+}$) is considered as the most toxic element that extensively present in the environment. The deposition of combustion pollutants into the atmosphere, mining, and the use of cadmium-containing fertilizers are significant anthropogenic sources of cadmium ions. Extensive exposure to cadmium can cause lung and breast cancer, heart failure, and abnormalities in glucose metabolism (*Alyasi et al., 2020*). According to the World Health Organization (*WHO, 2011*), the recommended level of cadmium in drinking water is three µg/L, while the highest allowable limit of cadmium ions in freshwater is 1.8 µg/L, according to the United States Environmental Protection Agency (*US Environmental Protection Agency, 2016*). Thus, an effective technology is desperately needed to eliminate excess cadmium in water to preserve human health and water quality.

Heavy metals are usually removed from wastewater using biotechnological and physicochemical methods, such as ion exchange, adsorption, electrochemical methods, chemical precipitation, flocculation/coagulation, and ion flotation. However, these methods have some disadvantages, such as high costs and disposal issues (*El-Desouky et al., 2021*; *Karami et al., 2017*).

Biosorption using living and dead microorganisms, such as micro- and macro-algae, fungi, bacteria, and agro-industrial wastes has become a very viable, economical, and ecologically benign alternative technique (*Fawzy et al., 2024*). Biosorption by macroalgae has many benefits, including high sensitivity and selectivity for particular heavy metals, high adsorption capacity and removal efficiency of heavy metals, low investment and operating costs, sustainability of biomass, and worldwide availability (*Aloufi et al., 2024*; *Selvakumar et al., 2023*). The high capacity of macroalgae to eliminate heavy metals is attributed to the composition of their cell walls, which include polysaccharides, lipids, and proteins. These components have various functional groups, such as hydroxyl, sulfate, carboxyl, amino, carbonyl, and others, which can act as effective ion exchangers (*Arumugam et al., 2018*). Regarding the red algae used in the current study, their cell wall is composed of cellulose, but their biosorption capacity is primarily due to the presence of sulfated polysaccharides (*Anastopoulos & Kyzas, 2015*).

Many macroalgae, including *Turbinaria ornate* (*Fawzy et al., 2022a*), *Cystoseria indica* (*Khajavian et al., 2019*), *Ulva fasciata* (*El-Naggar et al., 2018*), and *Durvillaea antarctica* (*Gutiérrez et al., 2015*), have shown the ability to effectively eliminate cadmium ions from

wastewater. Biosorbents can also be modified or pretreated with alkali, acid and calcium chloride in order to improve their characteristics by inhibiting their leaching and raising the amount of active sites on the surface (*Plazinski, 2013*).

To the best of our understanding, this is the first study that discusses the removal of cadmium ions from wastewater using native red macroalgae *Digenia simplex* pretreated with calcium chloride. Therefore, the aim of this study was to remove cadmium from aquatic environments using chemically modified *D. simplex*. The important factors affecting the biosorption of cadmium, such as pH, metal concentration, temperature, contact time and biosorbent dosage were investigated. Response surface methodology (RSM) was subsequently used to evaluate the optimal removal circumstances. The behavior of the cadmium biosorption on the pretreated algal biomass was also examined using kinetics, isotherms, and thermodynamics. In addition, the pretreated biosorbent was characterized using Fourier transform infrared spectroscopy (FT-IR), scanning electron microscope (SEM), and energy-dispersive X-ray spectroscopy (EDX). Finally, the optimal conditions determined using RSM were applied to eliminate cadmium from real agricultural wastewater collected from Taif, Saudi Arabia.

## MATERIALS & METHODS

### Algal biomass preparation

Samples of the native red marine macroalga *Digenia simplex* (Wulfen) C. Agardh were obtained from the Red Sea shores in Jeddah, Saudi Arabia. The samples were washed with deionized water in order to get rid of epiphytes, sand and impurities. The algae were then dried in an oven at 45 °C, ground into particles of 100–200 μm, and placed in air tight bottles at room temperature until required.

### Pretreatment with calcium chloride

Dried *D. simplex* samples were treated with 0.2% calcium chloride ($CaCl_2$) at a ratio of 1 g:100 ml to improve the biosorption capacity of the algal biomass and then stirred with a magnetic stirrer at 25 °C for 2 h. Subsequently, the algae samples pretreated with $CaCl_2$ (DSC) were filtered, washed with deionized water, and dried in an oven at 45 °C until their weight remained constant (*Omar, El-Gendy & Al-Ahmary, 2018*).

### Algal biomass characterization

*Digenia* algae pretreated with $CaCl_2$ were characterized before and after cadmium biosorption using two instrumental techniques, one of them is FTIR spectrometry (FTS, 3000 MX; Bio-Rad, Hercules, CA, USA), which is helpful in determining the type of functional groups present on the surface of the biosorbent. For sample preparation, a certain amount of biosorbent was thoroughly mixed with 0.4 g of KBr, and the mixture was analyzed in the spectral range of 4,000–400 $cm^{-1}$. The other technique is scanning electron microscopy (SEM, JSM-5400 LV; JEOL), attached to energy-dispersive X-ray spectroscopy (EDX, JSM-IT 200; JEOL), which enables the examination of morphological structure and elemental composition of the algae surface. EDX analysis was carried out using a silicon drift detector (SDD) at a working distance of 15 mm, an accelerating voltage of 20 kV, and live acquisition duration of 60 s.

## Biosorption studies

Batch biosorption investigations were performed in 250 mL conical flasks containing 100 mL of distilled water and the required cadmium concentration. Appropriate amounts of $CdSO_4.8/3H_2O$ were dissolved in 1,000 mL of deionized water for preparing a stock solution of cadmium with varying concentrations. In this study, different independent variables at different levels were used in order to maximize the $Cd^{2+}$ ion removal efficiency. These factors included exposure time between 0 and 150 min, $Cd^{2+}$ ion concentrations between 10 and 50 mg/L, and temperature of 25, 35, and 45 °C. The factors to be examined were varied while retaining the other conditions (pH 6, Cd concentration 30 mg/l, pretreated biomass concentration five g/l, and exposure time 60 min). The solutions were stirred at a fixed temperature of 25 °C and a speed of 170 rpm. The initial pH was adjusted with NaOH and/or $H_2SO_4$ (0.1 M). Each trial was repeated three times. After the experiment was completed, the algae biomass was removed from the aqueous solution by centrifugation at 4,000 rpm for 5 min. The cadmium concentration in the solution was then evaluated using an atomic absorption spectrometer (Buck Scientific 210VGP, Inc., East Norwalk, CT, USA).

The cadmium removal percentage (R, %) and biosorption capacity ($q_e$) were calculated from Eqs. (1) and (2) as follows:

$$q_e \ (mg/g) = \frac{V(C_0 - C_f)}{X_m} \tag{1}$$

$$(R \ \%) = \frac{C_0 - C_f}{C_0} \times 100 \tag{2}$$

where $C_0$ is the initial $Cd^{2+}$ concentration (mg/L), $C_f$ is the final $Cd^{2+}$ concentration at any time (mg/L), $X_m$ is the weight of algal biomass (g/L), and V is the volume of $Cd^{2+}$ solution (mL).

## Experimental design using RSM and BBD

The combined influences of different factors on the cadmium removal percentage by DSC were tested and modeled by RSM involving a Box–Behnken Design (BBD). The independent variables in the design were pH (3, 6, 9), initial cadmium concentration (10, 30, 50 mg/L), and algal dose (one, five, nine g/L). The quadratic model consisted of 15 experiments with three replicates to quantify experimental error. The BBD used a three-level design with test parameters at low, medium, and high (0, −1, and +1, respectively). The experiment was designed using Stat-Ease design expert v7.0 (Stat-Ease, Minneapolis, MN, USA). The experiment was conducted at a constant temperature of 25 °C and a contact time of 60 min, and the remaining cadmium concentration was assessed as earlier described.

The impacts of each independent parameter and their interactions were analyzed using the following second-order equation:

$$Y = \beta_0 + \sum \beta_i X_i + \sum \beta_{ii} X_{i2} + \sum \beta_{ij} X_i X_j \tag{3}$$

where $Y$ is the expected $Cd^{2+}$ removal, $X_i$ and $X_j$ describe the studied parameters, and $\beta_0$ represents the intercept. $\beta_i$, $\beta_{ii}$, and $\beta_{ij}$ are the linear, quadratic, and interaction coefficients, respectively.

## Validation of predictive model and optimized conditions

Derringer's desirability function approach was used to identify the optimal conditions of cadmium removal efficiency. Three experiments were carried out under these circumstances. The optimization was confirmed at an optimal pH of 5.78, an initial Cd concentration of 24.79 mg/L, and an algal dose of 6.13 g/L, with a contact time of 60 min at a temperature of 25 °C. The applicability and accuracy of the quadratic model were determined by comparing the experimental results with the expected results.

## Kinetics, isotherms, and thermodynamic biosorption studies

For the kinetic experiments, 100 mL of cadmium solution (30 mg/L) was mixed with five g/L macroalgae concentration (DSC) in a 250 mL conical flask at 25 °C and pH 6. The mixture was then shaken at 170 rpm on a rotary shaker. Samples were taken from the cadmium-containing solution at intervals of 0–150 min, filtered, and the cadmium concentration was determined as mentioned previously. The kinetics of cadmium adsorption using $CaCl_2$-pretreated *Digenia* were analyzed by intraparticle diffusion (Eq. (4)), pseudo-first-order (Eq. (5)), and pseudo-second- order (Eq. (6)) as follows:

$$q_t = K_i t^{0.5} + C_i \tag{4}$$

$$\text{Log}(q_e - q_t) = \text{Log}q_e - \frac{K_1 t}{2.303} \tag{5}$$

$$\frac{t}{q_t} = \frac{1}{K_2 q_e^2} + \frac{t}{q_e} \tag{6}$$

where $q_t$ (mg/g) is the amount of cadmium adsorbed on the algae surface at any time (t), $C_i$ is the boundary thickness, $K_i$ (mg/g min$^{1/2}$), $K_1$ (1/min) and $K_2$ (g/mg min) are the rate constants of intra-particle, pseudo-1st -order, and pseudo-2nd-order, respectively.

For isothermal experiments, five g/L algal dosage (DSC) was added to 100 ml of cadmium solution with concentrations ranging from 10 to 50 mg/L, pH 6, and temperature 25 °C for 60 min. The mixture was shaken at 170 rpm, and then the residual cadmium concentration was determined. In order to evaluate the most appropriate biosorption isotherm for the biosorption of cadmium on the pretreatment algae, the Freundlich, Langmuir, and Dubinin–Radushkevich isotherm models were calculated as follows:

Freundlich model:

$$\ln q_e = \ln K_f + \frac{1}{n} \ln C_{eq} \tag{7}$$

Langmuir isotherm model:

$$\frac{C_{eq}}{q_e} = \frac{1}{q_{max} b} + \frac{C_{eq}}{q_{max}} \tag{8}$$

$$R_L = 1/(1 + bC_o) \tag{9}$$

Dubinin–Radushkevich isotherm model:

$$\ln q_e = \ln q_0 - \beta \varepsilon^2 \tag{10}$$

$$\varepsilon = RT\ln\left(1 + \frac{1}{C_{eq}}\right) \tag{11}$$

$$E = \sqrt{1/2\beta} \tag{12}$$

where $C_{eq}$ (mg/L) represents the equilibrium metal concentration after the biosorption process, n and $K_f$ are Freundlich constants, b and $q_{max}$ represent the affinity constant and maximum adsorption capacity of the Langmuir isotherm model, respectively, $R_L$ is the dimensionless separation factor, $C_0$ (mg/L) represents the initial concentration of $Cd^{2+}$ ions. $\beta$ ($mol^2/J^2$) and $q_0$ (mg/g) are the mean biosorption energy and the biosorption capacity of Dubinin–Radushkevich, respectively, $\varepsilon$ is the Polanyi potential, T (K) and R (8.314 kJ/mol) are the absolute temperature and universal gas constant, respectively, and E (kJ/mol) represents the average free energy of biosorption.

Finally, thermodynamic investigations were carried out at different temperatures (25, 35, and 45 °C). For this purpose, five g/L DSC was added to 100 ml of 30 mg/L cadmium solution (pH 6) and stirred for 60 min. The cadmium concentration was then measured as earlier described. Thermodynamic parameters were calculated using the following equations:

$$K_C = \frac{Cs}{C_{eq}} \tag{13}$$

$$lnK_C = \frac{\Delta S^o}{R} - \frac{\Delta H^o}{RT} \tag{14}$$

$$\Delta G^o = \Delta H^o - T\Delta S^o \tag{15}$$

where Kc and Cs are the equilibrium constant and equilibrium cadmium concentration in the algal biomass, respectively. $\Delta G^o$, $\Delta S^o$, and $\Delta H^o$ are the free energy change, entropy change, and enthalpy change of biosorption, respectively.

### Removal of cadmium from wastewater in Wady Al-Arj district

The real wastewater sample was obtained from the Wady Al-Arj district of Taif, Saudi Arabia. The temperature, pH, total dissolved solids (TDS), electrical conductivity (EC), and $Cd^{2+}$ ion concentration of the sample were evaluated. The biosorption experiment was carried out under the optimal conditions according to the Box–Behnken Design (BBD). As the concentration of cadmium ions was found in a low amount, a definite concentration of $Cd^{2+}$ was added to the effluent to adjust the initial concentration of $Cd^{2+}$ at 24.79 mg/L which was determined by BBD.

### Statistical analysis

The experimental data were analyzed using Stat-Ease Design Expert v7.0. Multiple linear regression was used to calculate the regression coefficients for linear and quadratic parameters and term interactions. The statistical significance of each regression coefficient was determined by computing the $t$-value of the pure error replicates at the center point. The statistical evaluation of the quadratic model was performed using analysis of variance (ANOVA).

## RESULTS AND DISCUSSION

### Surface characterization

#### FT-IR analysis

FTIR analysis of *Digenia* pretreated with $CaCl_2$ was performed to estimate the type and number of functional groups on the algae surface involved in $Cd^{2+}$ biosorption. FTIR spectra of *Digenia* algae before and after cadmium biosorption revealed a considerable number of functional groups on the algae surface that can bind $Cd^{2+}$ ions in aqueous solution (Figs. 1A, 1B).

The peak observed at 3,424 $cm^{-1}$ is associated with the vibrations of N–H bonds, O–H bonds, and hydrogen bonds linking to amide group, showing the existence of proteins within the algae structure. This validates the role of hydrogen bonding in the biosorption process (*Oke & Mohan, 2022*). Other bands found at 2,923–2,927 $cm^{-1}$ can be associated with -CH bending vibrations of $CH_2$ and $CH_3$ groups (*Smith, 2021*). Furthermore, the bands observed at 1,750–1,747 $cm^{-1}$ before and after cadmium biosorption by *D. simplex* are connected with carboxylate ions (-COO-) (*Madadgar et al., 2023*). The peaks at 1,643–1,640 $cm^{-1}$ can be associated with the bending vibrations of the C=O bonds in the carboxylic and carboxyl functional groups, showing their participation in the biosorption process owing to electrostatic attraction (*Oke & Mohan, 2022*).

Some peaks appeared only after the biosorption of cadmium ions on the biosorbent surface, and significant changes in the intensity and position of the bands were detected in the algal biomass. These alterations confirmed the biosorption of cadmium by the chemically modified *D. simplex*. After Cd adsorption, the band in the biosorbent altered from 1,034 $cm^{-1}$ to 1,073 $cm^{-1}$. This shift shows the effective participation of CO groups in the biosorption process (*Ciobanu et al., 2023*). The bands at 1,160–1,153 $cm^{-1}$ before and after $Cd^{2+}$ biosoprtion showed the presence of S=O groups. In addition, the biosorption of cadmium changed the intensities of the absorption bands at 467 $cm^{-1}$, 533 $cm^{-1}$, 713 $cm^{-1}$, 858 $cm^{-1}$, and 875 $cm^{-1}$ to 466 $cm^{-1}$, 493 $cm^{-1}$, 712 $cm^{-1}$, 857 $cm^{-1}$, and 874 $cm^{-1}$, respectively. These changes reflect the interaction of $Cd^{2+}$ ions with *D. simplex*. The absorption bands at the wavelength 1,469–1,470 $cm^{-1}$ (O–H stretching vibration) are due to the coordinated $H_2O$ molecules and the new peak at 1,252 $cm^{-1}$ (O–H stretching vibration) observed following Cd biosorption shows the O–H groups on the algae surface (*Saberzadeh Sarvestani, Esmaeili & Ramavandi, 2016*). After the biosorption process, a new weak absorption band at 770 $cm^{-1}$ was observed due to the N–H stretching vibration of primary and secondary amines (*Wang et al., 2013*). According to *Lu et al. (2014)*, the peaks at 800 $cm^{-1}$ and 400 $cm^{-1}$ correspond to metal oxides ($CdO_2$).

The presence of these groups on the surface of the pretreated *Digenia* biomass is likely to provide high ion exchange capacity to the biosorbent (*Mohseni-Bandpei, Ramavandi & Kafaei, 2019*). Additionally, the decrease in band intensities following $Cd^{2+}$ biosorption on the surface of biosorbent demonstrates that the aforementioned functional groups are effective in the biosorption process.

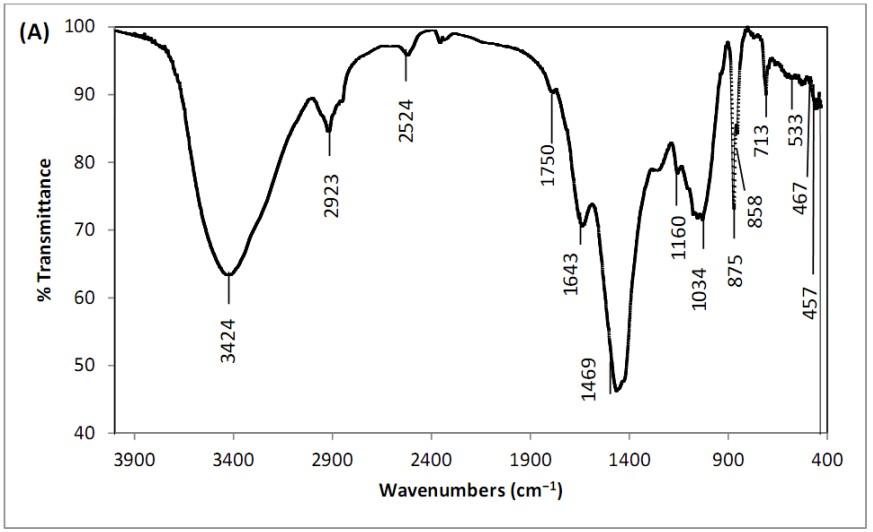

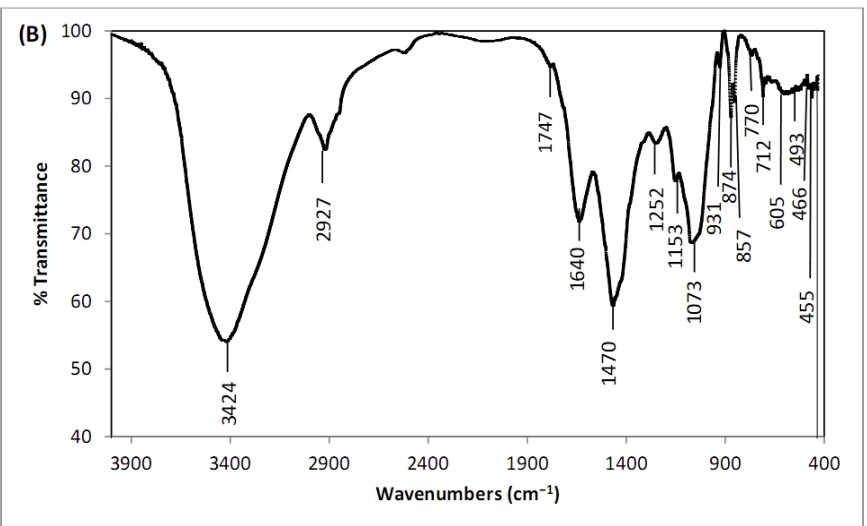

**Figure 1** FTIR of pretreated *Digenia* biomass (A) prior and (B) after cadmium biosorption.

### Analyses of SEM-EDX

SEM and EDX analyses were used to investigate the surface morphology and elemental alignment of the DSC biosorbent before and after cadmium biosorption and provide important information about the impact of adsorption on pretreated *D. simplex*. Significant alterations in the surface morphology of the algal biomass could be detected (Figs. 2A, 2B). The SEM image of the chemically modified *D. simplex* biomass (Fig. 2A) indicated the cell surface's porous structure. This structure increased the sorbent's surface area, hence facilitating the biosorption of cadmium. The presence of smooth pores with a flat shape after the biosorption process could be due to the accumulation of Cd ions on the surface of the pretreated *Digenia* biomass caused by electrostatic interactions (Fig. 2B) (*Isam et al.,*
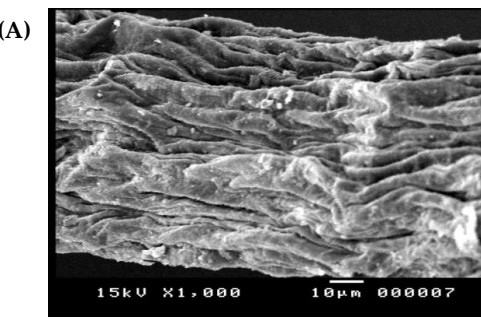 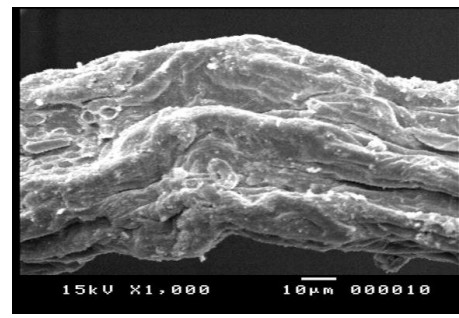

**Figure 2** Scanning electron micrographs of pretreated *Digenia* biomass (A) prior and (B) after Cd²⁺ ions biosorption.

*2023*). Our findings also demonstrated that some changes in the morphology of the algal biomass occurred after Cd biosorption.

The elemental composition of the DSC biomass before and after cadmium biosorption was examined by EDX spectroscopy. The algal biomass was found to contain carbon, oxygen, sodium, magnesium, silicon, sulphur, chloride, calcium, and cadmium (Figs. 3A, 3B). The appearance of the cadmium band in the spectrum indicates that algal biomass efficiently collects metal ions from the aqueous solution (Fig. 3B). Furthermore, the decrease and/or disappearance of sodium, potassium, chloride, and magnesium in the analysis may be due to the complexation and/or ion exchange of Cd²⁺ with monovalent or divalent cations of the algal biomass, which are liberated by Cd²⁺ (*Fawzy et al., 2022b*). *Aloufi et al. (2024)* and *Michalak, Mironiuk & Marycz (2018)* described similar biosorption mechanisms.

## Batch adsorption studies
### Effect of exposure time

Contact time is a key parameter affecting the biosorption process. Figure 4A displays the impact of exposure time on the removal efficiency (%), and biosorption capacity of Cd ions. Algal biomass is distinguished by its rapid biosorption of cadmium from aqueous solutions. In Fig. 4A, it can be noticed that after 10 min of treatment time, more than 97% of Cd²⁺ ions was biosorbed on the pretreated *D. simplex* biomass, which is attributed to the strong concentration gradient and the availability of additional free active sites on the surface of the pretreated *Digenia* biomass. Moreover, a quick equilibrium time of approximately 20 min was achieved. These findings are significant since equilibrium time is a major factor for cost-effective wastewater treatment applications (*Al-Zaban et al., 2022*). After this equilibrium time, the biosorption efficiency of Cd decreases or relatively uniform biosorption occurs due to the decrease of biosorption sites on the algal surface (Fig. 4A) (*Jabri et al., 2023*). The equilibrium time recorded in this study is shorter than that commonly observed for cadmium adsorption on other materials. Several investigations on biosorption showed that most algae achieved maximum ion removal within times ranging from 30 to 90 min (*Ibrahim, 2011; Amro & Abhary, 2019; Sayadi, Rashki & Shahri, 2019*).

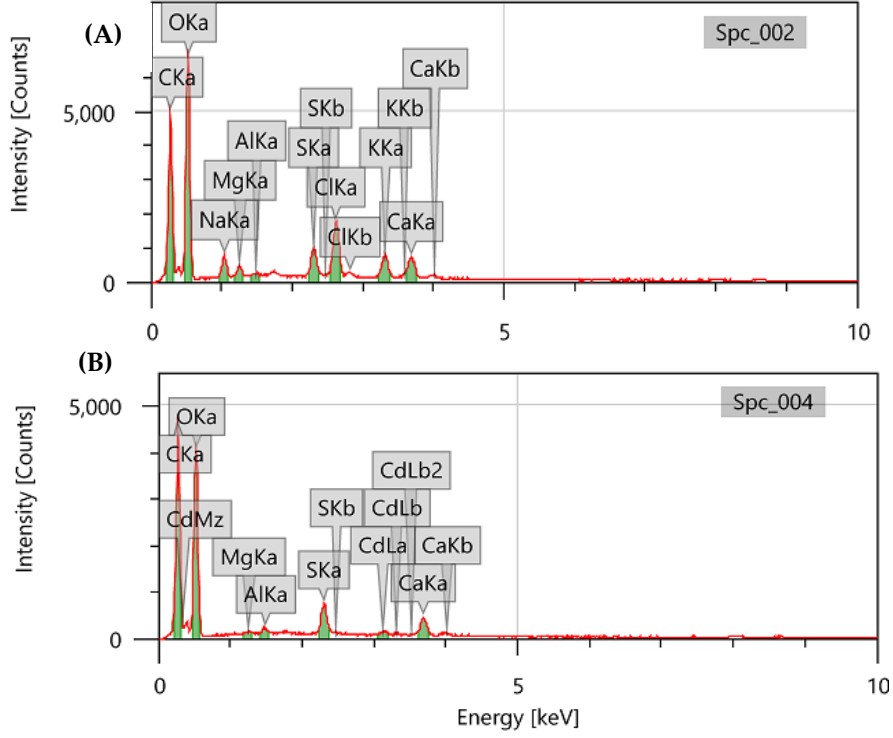

**Figure 3** EDX spectra of pretreated *Digenia* biomass (A) prior and (B) after cadmium biosorption.

As a result, the agitation period for the other biosorption experiments was set at 60 min to ensure complete equilibrium.

### Impact of initial cadmium concentration on biosorption process

Metal concentration is one of the significant parameters impacting the biosorbent's adsorption efficiency. As demonstrated in Fig. 4B, increasing the initial concentration of $Cd^{2+}$ ions from 10 mg/L to 50 mg/L enhanced the biosorption capacity of *D. simplex* biomass from 1.98 mg/g to 9.40 mg/g. However, the removal efficiency reduced from 98.8 to 94.0%. Consequently, the highest biosorption uptake of algal biomass occurred at a $Cd^{2+}$ concentration of 10 mg/L. The high cadmium removal rate at lower concentrations can be attributed to the presence of free functional groups on the algal surface such as hydroxyl, carboxyl, and amide groups, resulting in high cadmium removal efficiency (*Li et al., 2023*). However, with the increase in the initial cadmium concentration, the accessibility of these binding sites became saturated, resulting in an increase in the biosorption capacity (*Lu et al., 2020*).

### Impact of temperature on biosorption process

Temperature is another main factor to consider when assessing the removal of heavy metals from wastewater. The data revealed that temperature had slight impact on the biosorption capacity and removal efficiency of cadmium ions. Figure 4C shows that cadmium biosorption by pretreated *D. simplex* increased slightly with increasing temperature. The

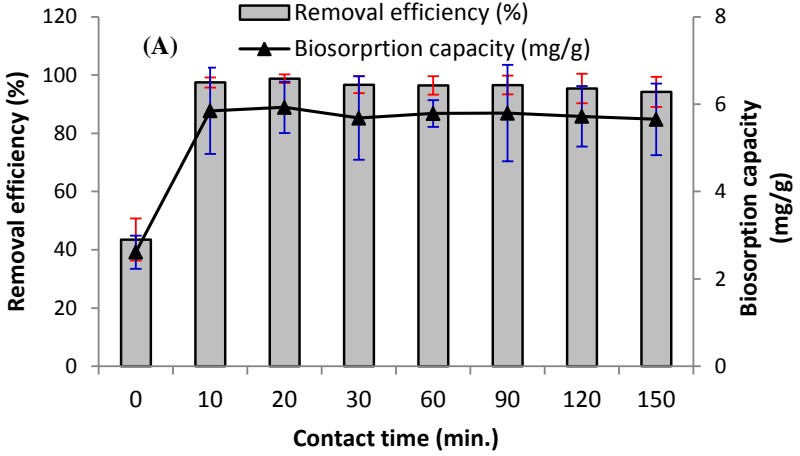

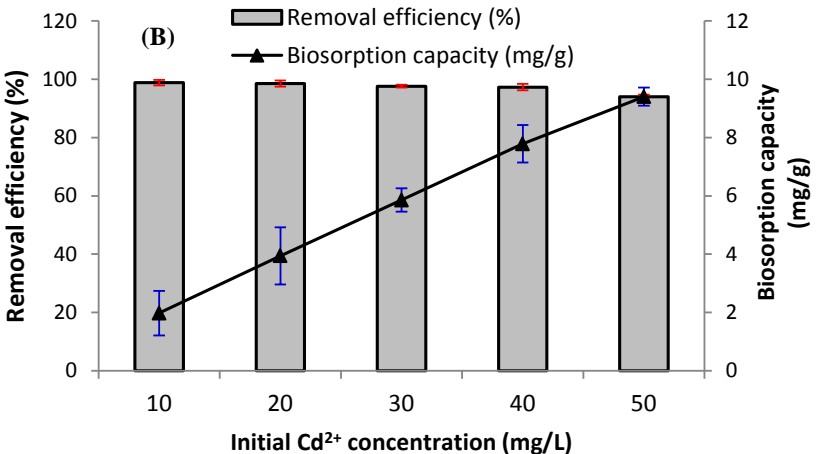

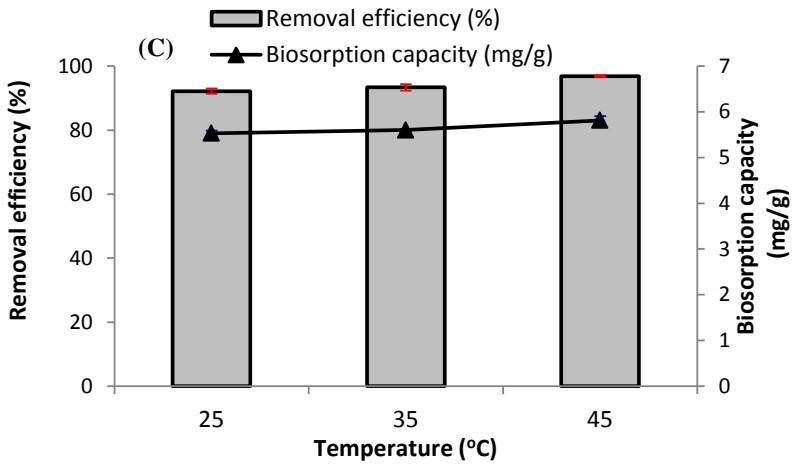

**Figure 4** Impact of (A) contact time, (B) initial cadmium concentration, and (C) temperature on the removal efficiency (%), and biosorption capacity of $Cd^{2+}$ ions ($q_e$) (biosorbent dose = 5 g/L, initial $Cd^{2+}$ concentration = 30.

higher biosorption at high temperatures could be attributed to the increase in the kinetic energy of $Cd^{2+}$ ions and the activity of algae surface, which makes it easier for cadmium to interact with binding sites on the surface, hence enhancing the biosorption of $Cd^{2+}$. Higher temperatures have also been demonstrated to accelerate the adsorption process by increasing the diffusion of driving force through the boundary layer and the diffusion rate within the adsorbent (*Rajasimman & Murugaiyan, 2021*). The fact that biosorption uptake of *D. simplex* biomass increases with increasing temperature proposes that the biosorption process is endothermic. Nevertheless, the minor changes in the biosorption capacity and removal efficiency values at relatively high temperatures (45 °C) indicate that it is not always advantageous to carry out the biosorption process at high temperatures. Accordingly, biosorption of cadmium can be performed at low temperature (25 °C), which is an efficient, cost-effective, and easy to replicate.

## Analysis of Box–Behnken design
### Evaluation of polynomial quadratic model and statistical analysis

The Box–Behnken design (BBD) is a statistical method used to design, model, and optimize cadmium removal by pretreated *D. simplex* (DSC). In this investigation, 15 runs were carried out to explore the impacts of three independent parameters such as pH, cadmium concentration, and algae dose on cadmium removal by *D. simplex* biomass. Table 1 shows that the percentage of $Cd^{2+}$ removal ranged from 81.55% to 99.48% (93.84% on average). The highest $Cd^{2+}$ biosorption percentage of pretreated *D. simplex* biomass was greater than the value reported by *Fawzy et al. (2022a)* for the maximum cadmium removal.

The second-order quadratic equation for cadmium removal efficiency by algal biomass is presented as follows:

$$\% \text{ Cd removal} = 98.88042 - 1.15A - 2.15B + 3.72C + 5.17BC - 3.13A^2 - 6.32C^2 \qquad (16)$$

where A, B, and C represent pH, $Cd^{2+}$ concentration, and algal dosage, respectively.

In this study, ANOVA was performed to examine the relationship between the studied factors and their impacts on the efficiency of cadmium removal by the biosorbent (*Alsawalha, 2023*). The data in Table 2 show that the quadratic RSM model was statistically significant with Prob. > $F$-value less than 0.05. The value of lack of fit was above 0.05, demonstrating that the model for Cd removal efficiency was also statistically significant (Table 2). Furthermore, the $R^2$ (0.90) and adjusted $R^2$ (0.83) values were close to each other, indicating a strong correlation between actual and expected values.

The data of this investigation were also analyzed to determine the suitability of the polynomial quadratic model. Figure 5A depicts a diagnostic scatter plot of the studentized residuals against the expected values for $Cd^{2+}$ adsorption on algal biomass. The random distribution of the residuals for the expected values of $Cd^{2+}$ biosorption along the central line shows that the actual values are consistent with the expected values. Furthermore, the range of the data points is ±3, showing that the design does not require a response transformation (*Isam et al., 2023*). Plotting the experimental results against the expected data shows an appropriate distribution along the straight line, indicating a satisfactory correlation (Fig. 5B). Since a large number of points are relatively close or centered on the

**Table 1  Experimental design of RSM quadratic model.**

| Trial | Independent variables | | | Dependent variable |
|---|---|---|---|---|
| | A pH | B Metal conc. (mg/L) | C Algal dosage (g/L) | Removal efficiency of $Cd^{2+}$ (%) |
| 1 | 3 (−1) | 10 (−1) | 5 (0) | 98.76 |
| 2 | 9 (+1) | 10 (−1) | 5 (0) | 96.97 |
| 3 | 3 (−1) | 50 (+1) | 5 (0) | 93.85 |
| 4 | 9 (+1) | 50 (+1) | 5 (0) | 96.63 |
| 5 | 3 (−1) | 30 (0) | 1 (−1) | 87.13 |
| 6 | 9 (+1) | 30 (0) | 1 (−1) | 81.55 |
| 7 | 3 (−1) | 30 (0) | 9 (+1) | 95.23 |
| 8 | 9 (+1) | 30 (0) | 9 (+1) | 90.61 |
| 9 | 6 (0) | 10 (−1) | 1 (−1) | 98.35 |
| 10 | 6 (0) | 50 (+1) | 1 (−1) | 82.05 |
| 11 | 6 (0) | 10 (−1) | 9 (+1) | 94.33 |
| 12 | 6 (0) | 50 (+1) | 9 (+1) | 98.70 |
| 13 | 6 (0) | 30 (0) | 5 (0) | 99.48 |
| 14 | 6 (0) | 30 (0) | 5 (0) | 98.17 |
| 15 | 6 (0) | 30 (0) | 5 (0) | 95.80 |

**Table 2  ANOVA data of the RSM quadratic model for $Cd^{2+}$ removal by DSC biosorbent.**

| Model term | Sum of squares | df | Mean squares | F-value | p-value Prob. >F |
|---|---|---|---|---|---|
| Model | 440.45 | 6 | 73.41 | 12.42 | 0.0011 |
| A-pH | 10.59 | 1 | 10.59 | 1.79 | 0.2175 |
| B-metal conc. | 36.88 | 1 | 36.88 | 6.24 | 0.0371 |
| C-Algal dose | 110.97 | 1 | 110.97 | 18.78 | 0.0025 |
| BC | 106.78 | 1 | 106.78 | 18.07 | 0.0028 |
| $A^2$ | 36.33 | 1 | 36.33 | 6.15 | 0.0382 |
| $C^2$ | 148.50 | 1 | 148.50 | 25.13 | 0.0010 |
| Residual | 47.28 | 8 | 5.91 | – | – |
| Lack of fit | 40.33 | 6 | 6.72 | 1.94 | 0.3791 |
| Pure error | 6.95 | 2 | 3.47 | – | – |
| Correlation total | 487.73 | 14 | – | – | – |
| $R^2 = 0.90$ | Adj. $R^2 = 0.83$ | Pred. $R^2 = 0.62$ | Adequate precision = 10.71 | Variation coefficient % = 2.59 | Mean = 93.8 |

diagonal line, this validates the model's applicability. As a result, the findings indicate that the RSM model utilized in this study is capable of identifying appropriate and accurate information about the relationship between the actual and the expected data. Another diagnostic plot is shown in Fig. 5C, indicating that the entire run residuals range are between values +3.29 and −2, and no pattern traces are expected.

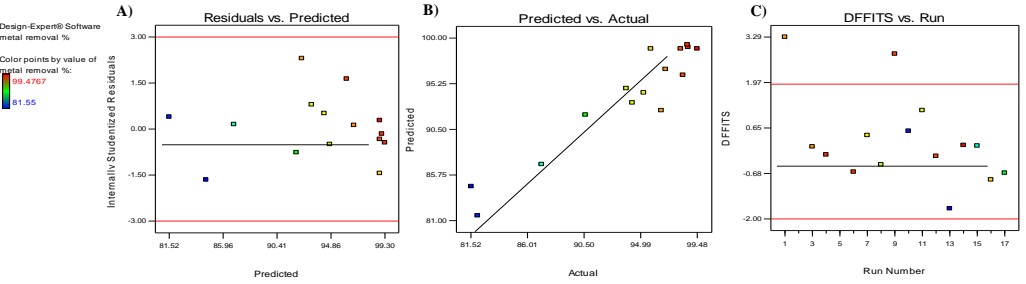

**Figure 5** (A) Internally studentized residual *versus* expected response plot, (B) expected removal efficiency against the actual efficiency plot, and (C) number of run for Cd²⁺ ion.

### Interpretation of RSM quadratic model

Response surface curves were plotted to better comprehend the correlation between the parameters and evaluate the optimal concentration of each parameter to obtain the maximum response. Figure 6A depicts the impact of simultaneous changes in pH and cadmium ion concentration on the efficiency of the biosorption behavior of chemically modified *D. simplex* in the uptake of cadmium using 3-D and contour graphs. The graphs show that pH alterations had no significant influence on the percentage of cadmium biosorption ($p > 0.05$; Table 2); but increasing the initial $Cd^{2+}$ concentration resulted in a significant decrease in the efficiency of cadmium removal by *D. simplex*. It is notable that the maximum removal efficiency was achieved at pH 6. The increase in $Cd^{2+}$ biosorption percentage at pH 6 can be attributed to the deprotonation of carboxyl, hydroxyl, and other negatively charged groups on the algal surface, which results in electrostatic interactions between the algal biomass and the positively charged $Cd^{2+}$ ions (*Ofudje et al., 2020*). FT-IR analysis confirmed these findings, which showed that cadmium biosorption primarily involves electrostatic interactions and ion exchange, and that the surface of *D. simplex* is composed of potential functional groups such as carbonyl, carboxyl, hydroxyl, and amide groups, which play a significant role in the interaction with cadmium ions. Similar maximum cadmium removal efficiencies at pH 6 was stated for *Cystoseria indica* (*Khajavian et al., 2019*), *Chlorella vulgaris* (*Goher et al., 2016*), and *Hypnea valentiae* (*Rathinam et al., 2010*).

Figure 6B shows the combined impacts of biomass dose and initial $Cd^{2+}$ level on the cadmium removal efficiency of *D. simplex*. As illustrated in the figure, the removal efficiency of cadmium ions decreased significantly from 97.84% to 84.78% with increasing initial $Cd^{2+}$ levels from 10 to 50 mg/L ($p = 0.037$; Table 2). The reason for this is the decrease in biosorption sites with increasing cadmium concentration for a fixed biomass of algae and increased intra-particle diffusion (*Asgari, Ramavandi & Farjadfard, 2013*). It is also worth mentioning that with increasing $Cd^{2+}$ concentration, the ratio of metal ions to algal biomass also increased, resulting in a decrease in Cd removal. The graph also shows that increasing the algae concentration from one to five g/l resulted in a relative increase in the removal efficiency from 94.57 to 97.84% (Fig. 6B). ANOVA indicated that the increase in metal uptake was statistically significant ($p = 0.002$; Table 2), which was attributed to the

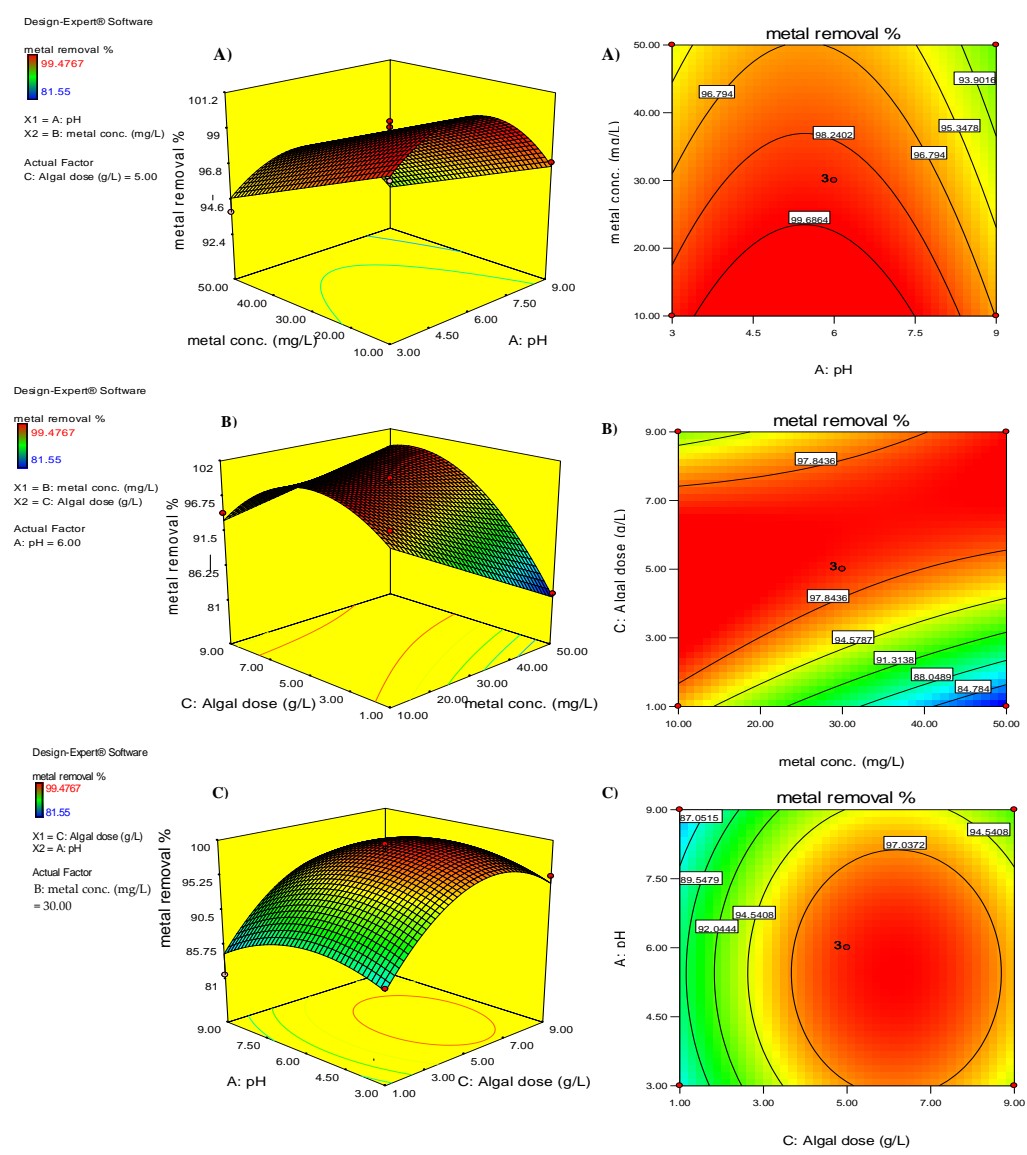

**Figure 6** The three-dimensional (left) and contour (right) graphs of Cd removal efficiency by pretreated *D. simplex* as a function of (A) pH and initial metal concentration, (B) initial Cd concentration and algal dose, (C) pH and algal dose.

increase in free binding sites on the algal surface, resulting in an increase in the elimination efficiency. When the biosorbent dose increased from five g/L to nine g/l, the percentage of cadmium biosorption decreased slightly from 97.84 to 91.31%. The reduction in the removal rate of cadmium at higher alga doses may be attributed to partial cell aggregation, which reduces the effective surface area available for the cadmium biosorption process (*Kaparapu & Krishna Prasad, 2018*). In this context, *Moawad, El-Sayed & El-Naggar (2020)* examined the $Cd^{2+}$ biosorption on different doses of *Cymodocea nodosa* biomass and found comparable results.

Figure 6C depicts the combined influence of pH and algae concentration on the biosorption percentage of cadmium by *D. simplex* biomass. When the initial concentration of cadmium was fixed at 30 mg/L, increasing pH and algal dose resulted in an increase in cadmium removal efficiency, followed by a decrease in removal efficiency within the experimental range. The joint impact of pH and algae biomass concentration was estimated as maximum Cd biosorption (97.03%) at pH 6 and algal dosage of five g/L.

### *Validation of the RSM model*

The optimization of the biosorption process was aimed at achieving the highest removal efficiency among all experiments tested under the examined circumstances. To optimize the cadmium removal efficiency of DSC biomass, the maximum desired target values of 81.52% to 99.30% (Table 1) were selected as the optimal values and the three parameters were adjusted within the studied range. The optimal removal efficiency was 97.27% at pH 5.78, initial $Cd^{2+}$ ion concentration of 24.79 mg/L, biomass dosage of 6.13 g/L and desirability function of 1.0. This corresponds to the expected value of 99.67%. Figure 7 depicts the optimal parameter ramps, confirming the predictability and accuracy of the proposed model.

## Evaluation of biosorption kinetics

Three kinetic models (intra-particle diffusion, pseudo-first and-second orders) were applied to examine the data and to estimate the best model that explains the mechanism of cadmium biosorption on the *D. simplex* surface.

In this study, the intra-particle diffusion kinetic model was applied to comprehend the role of the elementary diffusion stages in cadmium biosorption on *Digenia* biomass pretreated with calcium chloride. Figure S1 depicts the cadmium biosorption by DSC biosorbent, which can be divided into two stages. The first stage is characterized by diffusion of $Cd^{2+}$ ions on the alga's outer surface (film diffusion), whereas the second stage is characterized by cadmium ions diffusion *via* the internal pores of the biomass particles (intra-particle diffusion) (*Bouabidi et al., 2018*). As indicated in Table 3, the first sharper area displays an intra-particle diffusion rate constant ($k_{i1}$) of 0.332 mg/g min$^{1/2}$, related to quicker surface biosorption and external mass transfer. Nevertheless, the next gradual region indicates a slower diffusion rate with a $k_{i2}$ value of 0.0024 mg/g min$^{1/2}$, increasing the contact time needs to achieve equilibrium in the second step (Fig. S1A). It is clear that the elementary diffusion stage is significant in the biosorption process, but pore diffusion is not the rate-controlling stage for cadmium biosorption because the linear plot does not pass through the origin and the regression coefficient is low ($R^2 = 0.805$; Table 3). The deviation from the origin point can be attributed to the difference in mass transfer rates during the first and end stages of cadmium biosorption. This suggests a certain degree of boundary layer control and demonstrates that intraparticle diffusion is not the only rate-controlling stage. Chemical binding can also limit the biosorption rate, or both can occur concurrently. These findings are supported by the non-zero constant $C_i$ values (Table 3), showing that film diffusion on the outer algal surface happens simultaneously with diffusion inside the pores (*Fawzy et al., 2022a*). Furthermore, it was found that cadmium ions more easily bind

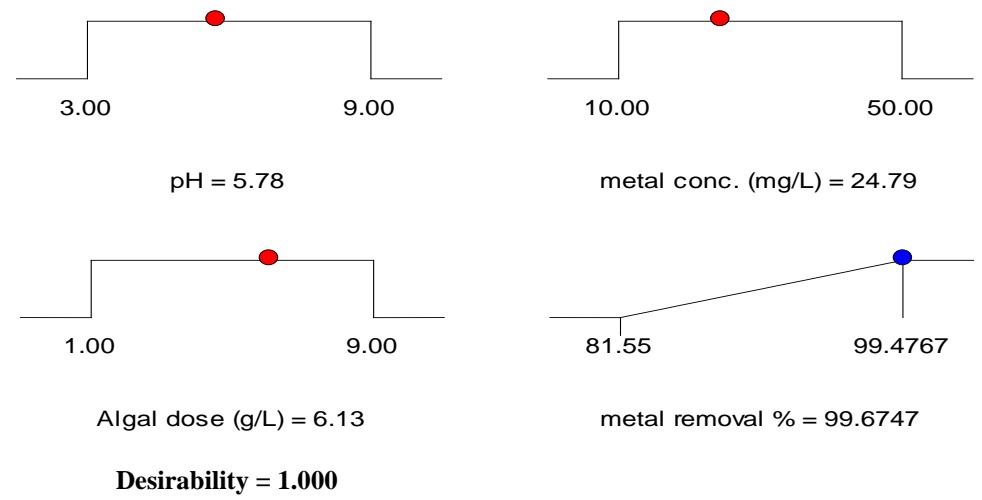

| | |
|---|---|
| 3.00 | 9.00 |
| pH = 5.78 | |

| | |
|---|---|
| 10.00 | 50.00 |
| metal conc. (mg/L) = 24.79 | |

| | |
|---|---|
| 1.00 | 9.00 |
| Algal dose (g/L) = 6.13 | |

| | |
|---|---|
| 81.55 | 99.4767 |
| metal removal % = 99.6747 | |

**Desirability = 1.000**

**Figure 7** Desirability ramp for the optimization of the independent factors and cadmium removal efficiency by pretreated *Digenia* biomass.

**Table 3** Kinetic models and their parameters for the biosorption of cadmium on DSC biosorbent.

| | Intra-particle diffusion | | | | | | Pseudo-1$^{st}$-order | | | Pseudo-2$^{nd}$-order | | |
|---|---|---|---|---|---|---|---|---|---|---|---|---|
| $q_e$ (exp.) (mg/g) | $K_{i1}$ (mg/g min$^{1/2}$) | $C_{i1}$ (mg/g) | $R_1^2$ | $K_{i2}$ (mg/g min$^{1/2}$) | $C_{i2}$ (mg/g) | $R_2^2$ | $q_e$ (cal.) (mg/g) | $K_1$ (min$^{-1}$) | $R^2$ | $q_e$ (cal.) (mg/g) | $K_2$ (g/mg min) | $R^2$ |
| 5.93 | 0.332 | 3.14 | 0.768 | 0.0024 | 5.858 | 0.805 | 1.432 | 0.033 | 0.658 | 5.672 | 0.264 | 0.999 |

to the biosorbent's functional groups. This reveals that the pretreated *D. simplex* biomass has a heterogeneous surface, as shown by SEM analysis, and the ease with which cadmium ions enter through the cracks on the algal surface depends primarily on its structural properties.

In the pseudo-first-order kinetic model, the reaction rate depends on the change between the adsorbate equilibrium concentration in the adsorbent's solid phase and the instantaneous concentration in the solid phase. Table 3 shows that the measured equilibrium value ($q_e$ exp. = 5.93) is inconsistent with the calculated value of the pseudo-first-order model ($q_e$ cal. = 1.432). In addition, the regression coefficient is low ($R^2 = 0.658$; Fig. S1B), showing that Cd$^{2+}$ biosorption on the pretreated *Digenia* biomass does not follow the pseudo-first-order model.

The pseudo-second-order model assumes that the rate-controlling phase is chemisorption. From Table 3 and Fig. S1C, it can be seen that the pseudo-second-order model with a regression coefficient close to 1.0 is in good agreement with the experimental results. Furthermore, the experimental $q_e$ value (5.93 mg/g) was found to be quite similar to the calculated $q_e$ value (5.672 mg/g), demonstrating that cadmium biosorption is controlled by chemical processes and indicates their charge specificity during equilibrium, such as covalent forces or electron exchange (*Sahoo & Prelot, 2020*).

From the previous data, it can be concluded that the pseudo-second-order model has higher regression coefficient value than the intra-particle diffusion model, indicating the presence of a chemisorption mechanism (*Ciobanu et al., 2023*). This validates that cadmium biosorption is a combination of surface chemical biosorption on the algal particle's boundary layer and intraparticle diffusion.

## Evaluation of biosorption isotherms

The adsorption isotherm is a significant parameter in the design of adsorption systems because it affects the adsorbent's capacity and the interaction of the adsorbate with the adsorbent (*Thomas et al., 2019*). In the current investigation, the biosorption mechanism was studied using Freundlich, Langmuir, and D–R adsorption isotherm models (*Khorshidi et al., 2016*).

The Freundlich model assumes that the adsorption of metal ions occurs on heterogeneous surfaces *via* multilayer adsorption, which is mainly dominated by complexation through lateral interactions between metal ions present on the adsorbent surface. The heterogeneous nature of the algal biomass is expressed in terms of adsorption intensity (n), which is calculated from the curve (ln $q_e$) against ln $C_{eq}$ (Fig. S2A). In this study, the 1/n value was within the range of $0 < 1/n < 1$ (0.48; Table 4). Moreover, the value of parameter n is higher than 1.0, showing that biosorption of cadmium is favored and there is a strong affinity between $Cd^{2+}$ ions and algal biomass, facilitating chemical biosorption. Due to the high regression coefficient ($R^2 = 0.947$), it can be concluded that the Freundlich model fits the data well and demonstrates the relationship between heterogeneous chemical multilayer adsorption and cadmium biosorption.

The Langmuir model assumes that the adsorption process occurs on a specific adsorption surface and that the attraction between ions reduces with increasing distance from the adsorption surface. It also indicates that the surface of biosorbent is homogenous. The regression coefficient for *D. simplex* was 0.997 (Table 4; Fig. S2B), demonstrating that the Langmuir model accurately predicted the cadmium biosorption on the pretreated *Digenia* biomass. Moreover, the calculated $q_{max}$ value (11.16 mg/g) was relatively consistent with the experimental biosorption capacity of *D. simplex* ($q_e = 9.40$ mg/g). This suggests that the biosorption process on the homogeneous surface occurs mainly *via* ion exchange, resulting in monolayer coverage of the pretreated *Digenia* biomass surface with cadmium ions.

Table 5 compares the Langmuir maximum adsorption capacity with the data from other $Cd^{2+}$ adsorption studies. It can be observed that the adsorption capacity of *D. simplex* is relatively comparable to or even greater than that of the other adsorbents. It can be concluded that DSC alga has a high potential for the removal of cadmium ions from wastewater. The structure of the red algae cell wall, composed of cellulose and sulfated polysaccharides, including carrageenan and agar, is responsible for this high capacity by binding metal ions to the algae biomass. Furthermore, the treatment of *D. simplex* with calcium chloride improved the binding sites for adsorption with cadmium ions (*Ordóñez et al., 2023*).

**Table 4 Equilibrium isotherm models and their parameters for the biosorption of cadmium on DSC biosorbent.**

| Freundlich | | | Langmuir | | | | Dubinin–Radushkevich | | | |
|---|---|---|---|---|---|---|---|---|---|---|
| $K_f$ (L/mg) | 1/n | $R^2$ | $q_{max}$ (mg/g) | b (L/mg) | $R_L$ | $R^2$ | $q_o$ (mg/g) | $\beta \times 10^{-8}$ (mol²/J²) | E (kJ/mol) | $R^2$ |
| 6.50 | 0.48 | 0.947 | 11.16 | 1.00 | 0.019–0.083 | 0.997 | 8.42 | 5.0 | 31.26 | 0.979 |

**Table 5 Comparison of the maximum biosorption capacities ($q_{max}$) of cadmium by various adsorbents.**

| Sorbents | Conditions | $q_{max}$ (mg/g) | Reference |
|---|---|---|---|
| Natural foxtail millet shell | Dose: 4 g/L, pH: 6.0, Time: 120 min, Temperature: 25 °C. | 12.48 | *Peng et al. (2018)* |
| Sawdust carbon ($Fe_3O_4$/SC) nanocomposite | Dose: 2 g/L, pH: 6.5, Time: 120 min, Temperature: 27 °C. | 22.0 | *Kataria & Garg (2018)* |
| Natural *Cladophora* sp. | Dose: 0.2 g/L, pH: 4.0, Time: 60 min, Temperature: 32 °C. | 12.07 | *Amro & Abhary (2019)* |
| Orange peel (KOH) | Dose: 80 g/L, pH: 5.0, Time: 60 min, Temperature: 20 °C. | 9.51 | *Romano et al. (2020)* |
| Rice husk (KOH) | Dose: 3 g/L, pH: 5.0, Time: 60 min, Temperature: 20 °C. | 8.49 | *Romano et al. (2020)* |
| Oil palm bagasse ($Al_2O_3$ nanoparticles) | Dose: 0.5 g/L, Time: 120 min, Temperature: 28 °C. | 17.40 | *Herrera-Barros et al. (2020)* |
| Natural Moroccan clay | pH: 5.0. | 5.12 | *Abbou et al. (2021)* |
| *Moringa oleifera* (FeNP/NFC) | Dose: 0.05 g/L, pH: 5.0, Temperature: 25 °C. | 12.5 | *Vázquez-Guerrero et al. (2021)* |
| Natural *Turbinaria ornata* | Dose: 5 g/L, pH: 5.0, Time: 90 min, Temperature: 25 °C. | 23.9 | *Fawzy et al. (2022a)* |
| *Digenea simplex* ($CaCl_2$) | Dose: 4 g/L, pH: 6.0, Time: 60 min, Temperature: 25 °C. | 11.16 | This study |

The dimensionless constant, termed the separation factor ($R_L$), describes the favorable biosorption process and is determined by the Langmuir isotherm constant (b) as described in Eq. (9). According to the value of the separation factor, the adsorption is favorable when $0 < R_L < 1$, linear when $R_L = 1$, unfavorable when $R_L > 1$, or irreversible when $R_L = 0$. The values of the separation factor in this investigation were smaller than 1.0 (0.019−0.083) (Table 4), indicating that the Langmuir model fits the biosorption of cadmium ions on *D. simplex* biomass well (*Abbou et al., 2021*).

In this study, the Dubinin–Radushkevich (D–R) model was examined to distinguish between chemical and physical adsorption (*Esvandi et al., 2020*; Fig. S2C). The value of mean free energy (E), which is commonly used to determine the type of adsorption, was 31.62 kJ/mol (Table 4), which is higher than 16 kJ/mol (*Aloufi et al., 2024*). This shows that the biosorption process of cadmium ions in nature can be described as chemical biosorption. This result verifies previous data obtained from the Freundlich model and pseudo-second-order model.

In general, the $R^2$ values obtained by the Freundlich, Langmuir, and D–R models were all higher than 0.94. This indicates that the three isotherm models are in good agreement

with the equilibrium results and verifies the efficiency of the pretreated *D. simplex* in eliminating cadmium ions from aqueous solutions.

## Thermodynamic analysis

Thermodynamic investigations of $Cd^{2+}$ biosorption on the surface of pretreated *D. simplex* were carried out at different temperatures in order to determine the resilience and feasibility of the adsorption process. Table 6 displays the values of the change in free energy ($\Delta G^o$), entropy change ($\Delta S^o$), and enthalpy change ($\Delta H^o$). As shown in the table, the biosorption of $Cd^{2+}$ ions at temperatures of 298, 308, and 318 K resulted in free energy changes of −2.12, −2.64, and −4.82 kJ/mol, respectively. The increase in temperature resulted in a reduction in the free energy change values, indicating a higher feasibility of cadmium biosorption (Fig. S3). Additionally, the negative $\Delta G^o$ values showed that the biosorption process of cadmium was feasible and spontaneous. Moreover, the positive value of $\Delta H^o$ showed that the endothermic class was responsible for the adsorption behavior of cadmium ions on the pretreated *Digenia* algae (Alharbi et al., 2022). This explains why the removal rate of cadmium ions increased with increasing temperature. According to Menezes et al. (2020), the values of $\Delta H^o$ between 80 and 200 kJ/mol, which represent the heat of chemical reaction, are commonly supposed to be comparable to values related to chemical adsorption processes. However, there is no specific standard associated with the values of $\Delta H^o$ to define the adsorption process. Therefore, the $\Delta H^o$ value (38.01 kJ/mol) in this investigation indicates that the $Cd^{2+}$ biosorption process occurs through chemical biosorption rather than physical adsorption, which is consistent with the observations obtained previously. Additionally, the interaction between the active binding sites on the algal biomass and the cadmium ions is electrostatic (ion-exchange type), which is supported by the $\Delta H^o$ and $\Delta G^o$ values and the positive and low value of $\Delta S^o$ (0.134 kJ/mol K; Table 6). The positive $\Delta S^o$ value shows the degree of disorderliness at the solid-solution interface during the biosorption process (Gorzin & Bahri Rasht Abadi, 2018). This phenomenon is typical of ion exchange processes, in which a binding of cadmium ion happens concurrently with the release of another one into the aqueous solution (Kadiri et al., 2019).

## Removal of cadmium from real wastewater

In this study, the removal of $Cd^{2+}$ ions from real wastewater was also investigated under optimal circumstances achieved from BBD (pH 5.78, initial $Cd^{2+}$ concentration 24.79 mg/L, and sorbent dosage 6.13 g/L), with a fixed contact time of 60 min and temperature of 25 °C. The characteristics of the effluent were evaluated as follows: temperature = 17 °C, pH of the solution = 8.74, TDS = 640.96 mg/L, and electrical conductivity = 1,001.5 µS/cm. Under the optimal conditions, the treatment of effluent with DSC biomass demonstrated high efficiency with 97.9% removal of $Cd^{2+}$ ions. Treatment with $CaCl_2$ enhances the surface characteristics of *D. simplex* biomass by stabilizing and activating functional groups such as hydroxyl, carboxyl, and amide groups. These groups play an important role in cadmium ion binding through complexation and ion exchange mechanisms (Li et al., 2023). These results show that *D. simplex* biomass can effectively remove $Cd^{2+}$ ions from wastewater.

**Table 6  Thermodynamic parameters of cadmium biosorption on DSC biosorbent.**

| Temperature (K) | $\Delta G^o$ (kJ/mol) | $\Delta H^o$ (kJ/mol) | $\Delta S^o$ (kJ/mol K) | $R^2$ |
|---|---|---|---|---|
| 298 | −2.12 | | | |
| 308 | −2.64 | 38.01 | 0.134 | 0.869 |
| 318 | −4.82 | | | |

## CONCLUSIONS

The marine red macroalga *D. simplex* was chemically modified with calcium chloride and investigated as an efficient biosorbent for the removal of cadmium from aqueous solutions. Individual factors, including contact time, $Cd^{2+}$ concentration, and temperature were examined. The biosorption kinetics followed pseudo-second-order model, but the biosorption isotherms followed the Freundlich, Langmuir and D–R models with a $q_{max}$ of 11.16 mg/g. Cadmium biosorption on pretreated *Digenia* biomass was also modeled and optimized using a statistical method by determining the influences of three variables, including pH, initial $Cd^{2+}$ level, and algal concentration. Numerical optimization revealed that the maximum $Cd^{2+}$ ion removal (97.27%) was achieved at pH 5.78, initial cadmium concentration 24.79 mg/L, and *D. simplex* dose (DSC) 6.13 g L, with a desirability of 1.0. The quadratic RSM model accurately predicted the removal of $Cd^{2+}$ as a lack of fit was not significant ($p > 0.05$) with a good regression coefficient value ($R^2 = 0.90$). According to the ANOVA data, algae dose and Cd concentration were the most important parameters for the elimination of cadmium by DSC. The optimal conditions derived by BBD were also applied to agricultural wastewater samples and the results showed that pretreatment of *Digenia* algae with $CaCl_2$ was effective in removing cadmium from wastewater. FT-IR, SEM and EDX analyses confirmed the interaction between several functional groups such as amide, carboxyl, hydroxyl and carbonyl groups on the algae surface and cadmium ions. Therefore, our study on a cost-effective biosorbent is extremely valuable for the development of sustainable biotechnologies for the removal of heavy metals from wastewater. Although this study showed high efficiency in cadmium removal, it did not investigate the efficacy of pretreated *D. simplex* as an economical and eco-friendly biosorbent for adsorbing other heavy metals and other pollutants. In addition, it did not employ additional modification or pretreatment techniques, which limits its applicability in other wastewater treatment applications. This study did not also investigate the efficiency of pretreated algae for large-scale removal of cadmium from real wastewater, as the optimization experiments were conducted in the laboratory on synthetic wastewater, which typically contains only one or a few metals. Therefore, further research is needed on different pre-treatment or modification techniques for algae, metal recovery, and biosorbent reuse. Future investigation should also focus on large-scale application of pre-treated biosorbents to treat real wastewater containing various heavy metals.

### Funding

This project was supported by Princess Nourah bint Abdulrahman University Researchers Supporting Project number (PNURSP2025R221), Princess Nourah bint Abdulrahman University, Riyadh, Saudi Arabia. This project was also supported by Sultan Qaboos University, Oman (Grant RF/SCI/BIOL/24/05). The funders had no role in study design, data collection and analysis, decision to publish, or preparation of the manuscript.

### Grant Disclosures

The following grant information was disclosed by the authors:
Princess Nourah bint Abdulrahman University, Riyadh, Saudi Arabia: PNURSP2025R221.
Sultan Qaboos University, Oman: RF/SCI/BIOL/24/05.

### Competing Interests

The authors declare there are no competing interests.

### Author Contributions

- Sedky H.A. Hassan analyzed the data, prepared figures and/or tables, authored or reviewed drafts of the article, and approved the final draft.
- Maryam M. Alomran conceived and designed the experiments, prepared figures and/or tables, authored or reviewed drafts of the article, and approved the final draft.
- Nada I.A. Alsugiran analyzed the data, authored or reviewed drafts of the article, and approved the final draft.
- Mostafa Koutb analyzed the data, prepared figures and/or tables, authored or reviewed drafts of the article, and approved the final draft.
- Hassan Ahmed analyzed the data, authored or reviewed drafts of the article, and approved the final draft.
- Mustafa A. Fawzy conceived and designed the experiments, performed the experiments, analyzed the data, prepared figures and/or tables, authored or reviewed drafts of the article, and approved the final draft.

### Data Availability

   Data is available in the Supplemental Files.

### Supplemental Information

Supplemental information for this article can be found online at http://dx.doi.org/10.7717/peerj.19776#supplemental-information.

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
