# Peer review of "Response surface optimization for cadmium biosorption onto the pre-treated biomass of red algae Digenia simplex as a sustainable indigenous biosorbent"

_PeerJ, doi:10.7717/peerj.19776_

## Round 0.1 · original submission · Major Revisions

The introduction’s structure should be revised. Please sure it is well-written without AI-generated text.

**Language Note:** The review process has identified that the English language must be improved. PeerJ can provide language editing services - please contact us at [email protected] for pricing (be sure to provide your manuscript number and title). Alternatively, you should make your own arrangements to improve the language quality and provide details in your response letter. – PeerJ Staff

Reviewer 1 ·

Basic reporting

• The manuscript would benefit from language revision. Ensure the correct use of punctuations, tense, and sentence structure, and measuring units to enhance readability and coherence. Avoid redundant presentation of information.
• Starting paragraphs of introduction lacks structural coherence and clarity, making it difficult to follow logical flow. Revise the background by clearly articulating specific problem and explicitly explaining how it impacts the environment and human health.
• Methodology lacks the details, making it difficult to reproduce the study. Provide precise descriptions of procedures, equipment, and materials, ensuring clarity and consistency throughout.

Experimental design

• How was drying performed? Was it sun-drying or over-drying?
• What were the storage conditions?
• Provide the experimental details of the pretreatment experiment? What was the temperature, processing conditions etc.?
• Description of all the equations should be presented in the “materials and methods” section.

Validity of the findings

• All the data has been provided, but some revisions are suggested for improvement of "results and discussion" section.
• This section would benefit from more descriptive and clear subheadings. Arrange results of one section in 1-2 major paragraphs.
• Preferably compare the results with studies from last 3-5 years.
• What was the control in this study? Did authors study the Cadmium removal potential of non-treated D. simplex? A comparison of these two would be beneficial to evaluate how much pretreatment contributed to the cadmium removal efficiency.

Additional comments

• Figure legends should be self-explanatory and descriptive.
• It would be better to combine impact analysis graphs in a single figure for better representation.

Reviewer 2 ·

Basic reporting

The keywords in the manuscript need to be revise according to research aims.
Introduction needs to be revise. Seems like written by some AI tools.
Aims and need of the study needs improvement.
All the figures need to be revised. The font style and font size are different and inconsistent.
Line 165: The temperature units are different from previously mentions. Make this units consistent for study.

Experimental design

Material methods section: Pretreatment with calcium chloride is not clear. The authors need to provide details about pretreatment such as time, temperature and other important factors which play an important role during pre-treatment.
The authors claimed to determined “Kinetics, isotherms, and thermodynamic biosorption studies” but didn’t mention how? Mentioned briefly.
Line 190: replace the word “real wastewater samples” with name of the wastewater you collected.
Line 191-196: What is BBD?
Which tool will you used for statistical analysis? Explain briefly in the manuscript.

Validity of the findings

Why didn’t authors characterized biomass before CaCl2 treatment?
Add the limitation and future aspect of this study based on your findings in conclusion section.
There are some paragraphs which are just 2-3 lines which needs to be restructure.
Revise table 3. Add pretreatment approaches for other biomass and most suitable condition for biosorption of cadmium. Also revise the title of table. Also add more information related to red algae or green algae for biosorption role.

Additional comments

The English language of the manuscript required extensive improvement as there are some grammar and typing errors found.

Reviewer 3 ·

Basic reporting

Manuscript entitled “Response surface optimization for cadmium biosorption onto the pre-treated biomass of red algae Digenia simplex as a sustainable indigenous biosorbent” submitted by Sedky H.A. Hassan, Maryam M. Alomran, Nada I. A. Alsugiran, Mostafa Koutb, Hassan Ahmed, Mustafa A. Fawzy, can be considered for publication in PEERJ-Journal, after a major revision.

Experimental design

1. Line 46: “…with entropy and enthalpy values of 0.134 and 38.01 kJ/mol, respectively” Check the measure units for these parameters.
2. Line 92: “Among the various types of macroalgae…”. This observation should be clearly reworded.
3. Line 126: “The Digenia algae pretreated with CaCl2 was characterized…”. Why this pretreatment was necessary??? This should be explained.
4. Line 195: Replace “resulting in a final concentration of 24.79 mg/L obtained from BBD” with “to adjust the initial concentration of Cd(II) at 24.79 mg/L”.

Validity of the findings

5. Line 254: “Our findings also demonstrated…”. This observation is hazardous and should be reworded.
6. Line 308: “It is also clear that the elementary diffusion stage…”. This observation should be clearly reworded.
7. Line 344: Replace “model has higher determination coefficient value” with “model has higher regression coefficient value”. Make this correction on entire manuscript.
8. Line 369, Eq. 7: Replace “Ln” with “ln”. Make this correction on entire manuscript.
9. Line 395: “…if not greater than, that of other adsorbents…”. Delete these references, because they are mentioned in Table 3.
10. Line 449, eqs. 13-15: All notations should be explained.
11. Line 464: “… between 20.9 and 418.4 kJ/mol…”. Please check again these values, because are too high.
12. Line 472: “The positive DS o value shows Cd2+ ions affinity…”. This is not true. Please correct this observation.
13. Analysis of Box-Behnken Design: This section should be moved after biosorbent characterization. Also, this section should be systematized. Pay attention on the aspects important for this study and delete general observations/comments.
14. Removal of cadmium from real wastewater: These observations should be supported by the experimental data.

---

## Round 0.2 · Minor Revisions

One of the reviewers have raised some minor questions!

Reviewer 1 ·

Basic reporting

--

Experimental design

--

Validity of the findings

--

Reviewer 2 ·

Basic reporting

Line 56: They kayword “Pre-treatment Digenia simplex” will be “Digenia simplex”.
Line 154: Remove word algae because DSC is already abbreviation of pretreated algae.

Experimental design

Line 111: The authors claim that red macroalgae were obtained from Red Sea shores. Did the authors collect it by themselves? And how do authors confirm about Digenia simplex?
Line 124-130: The authors describe FTIR, but the parameters at which the samples are run for FTIR and EDX are missing.
Please clarify whether the experiments were performed in a single run or with replicates. Additionally, error bars and standard deviation values should be included in the figures to reflect data variability.

Validity of the findings

Line 544: 3.7. Removal of cadmium from real wastewater: Based on your study, can you please provide a graphical presentation of cadmium removal? Also, improve the discussion of this section.

Reviewer 3 ·

Basic reporting

All my previous remarks and comments have been considered in this new version of the manuscript. In my opinion, the revised manuscript meets the criteria and can be published as original paper in PEERJ-Journal.

Experimental design

All my previous remarks and comments have been considered in this new version of the manuscript. In my opinion, the revised manuscript meets the criteria and can be published as original paper in PEERJ-Journal.

Validity of the findings

All my previous remarks and comments have been considered in this new version of the manuscript. In my opinion, the revised manuscript meets the criteria and can be published as original paper in PEERJ-Journal.

Additional comments

All my previous remarks and comments have been considered in this new version of the manuscript. In my opinion, the revised manuscript meets the criteria and can be published as original paper in PEERJ-Journal.

---

## Round 0.3 · accepted · Accept

Authors have addressed comments

**PeerJ Staff Note**: Although the Academic and Section Editors are happy to accept your article as being scientifically sound, a final check of the manuscript shows that it would benefit from further editing. Therefore, please identify necessary edits and address these while in proof stage. (Specifically, C. carnea and S. exigua should be consistently italicized.)

Reviewer 2 ·

Basic reporting

No further comments.

Experimental design

No further comments.

Validity of the findings

No further comments.

Additional comments

C. carnea and S. exigua should be consistently italicized. Kindly check them throughout the manuscript.